# Optimization of Water Plugging Characteristics and Mechanical Properties of Acrylate Grouting Materials Based on Composite Crosslinking Strategy

**DOI:** 10.3390/polym17060827

**Published:** 2025-03-20

**Authors:** Fengxian Yu, Langtian Qin, Deqiang Han, Feng Huang

**Affiliations:** 1Zhongke Jiantong Engineering Technology Co., Ltd., Beijing 101399, China; yufx@yuhong.com.cn (F.Y.); handq@yuhong.com.cn (D.H.); 2School of Engineering and Technology, China University of Geosciences Beijing, Beijing 100083, China; huangfeng@cugb.edu.cn

**Keywords:** acrylate grouting materials, composite crosslinking, mechanical properties, hydraulic stability, polymer network, anti-seepage

## Abstract

Traditional acrylate grouting materials often suffer from mechanical performance degradation and interfacial bonding failure under long-term water immersion, significantly limiting their application in pressurized water environments. This study proposes a composite crosslinking synergistic strategy to address these challenges. By constructing a dual-network structure through polyethylene glycol diacrylate (PEG500DA) and a monofunctional crosslinker (PEG-MA), and systematically optimizing the material formulation by regulating the triethanolamine content to control gelation time, the mechanical and hydraulic stability of the material was significantly enhanced. Increasing the acrylate concentration to 35% achieved an optimal balance between a slurry viscosity (8.3 mPa·s) and mechanical performance, with tensile strength reaching 76 kPa and the compressive strength of the sand-solidified body measuring 440 kPa. At a PEG500DA/PEG-MA ratio of 2:3, the material exhibited both high tensile strength (78 kPa) and exceptional ductility (elongation at break > 407%), with a compressive strength of 336 kPa for the sand-solidified body. When the total crosslinker content exceeded 5%, the 28-day water absorption and volume expansion rates were effectively reduced to 12% and 11%, respectively. Under simulated pressurized water conditions, the modified material demonstrated a water-pressure resistance of 300 kPa after 1 day, stabilizing at 350 kPa after 56 days—a 75% improvement over commercial products. This study provides an innovative solution for long-term anti-seepage applications in complex hydrogeological environments, offering significant advancements in material design and engineering reliability.

## 1. Introduction

Leakage problems in underground engineering structures seriously threaten engineering safety and service life, so the development of efficient seepage control materials has become a key technical demand in engineering construction [1,2,3,4,5]. As a new generation of chemical grouting materials, acrylate grouting materials form a polymer gel system with a three-dimensional network structure through the free radical copolymerization reaction of acrylate metal salt monomers and crosslinking agents. Compared with the traditional cement-based and epoxy-based grouting materials, acrylate grouting materials show significant technical advantages: their high reactivity enables rapid curing, low viscosity (2–15 mPa·s), with excellent crack permeability, and controllable gel time (from a few seconds to a few tens of minutes), which is able to adapt to the needs of different working conditions [6,7,8]. These characteristics make acrylate grouting materials play an important role in seepage and leakage control in water conservancy and hydropower, subway tunnels, underground pipeline corridors, and other major projects [9,10,11,12].

In recent years, the research on the performance enhancement of acrylate grouting materials mainly focuses on three directions: chemical molecular structure design, formulation composition optimization, and engineering adaptability improvement. In terms of molecular structure design, the mechanical properties and environmental stability of the materials have been significantly enhanced by introducing functional copolymerization monomers or modifying groups. For example, Zhang et al. [13] used in situ polymerization technology to graft acrylamide into an alumina sulfate cement matrix and constructed an organic–inorganic dual network structure, which increased the tensile strength of the material by 85%; Chen et al. [14] endowed the material with excellent water repellency by introducing fluorinated silane groups into the molecular chain of the epoxy acrylate. In the optimization of formulation composition, researchers focused on the synergistic effect between components and their influence on the material properties. Cui et al. [15] systematically studied the proportionality of monomers, crosslinking agents, initiators, and accelerators, and established a quantitative model for the gel time–component content; Shu et al. [16] optimized the fiber-polymer composite formulations by using orthogonal test methods to optimize the fluidity of the material and to reach an optimal balance of mechanical properties to achieve the best balance. In terms of engineering adaptability, Wang et al. [17] developed a high-permeability polymer material for the reinforcement of chalky sand strata and optimized its scour resistance by CFD simulation; Li et al. [18] designed a new grouting material with super-absorbent water, whose water-absorbent multiplication rate can reach 50–100 times its own weight in response to the demand for karst water surge management.

However, the existing research mainly focuses on a single modification mean, and the systematic understanding of the evolution of the performance of materials in complex environments is still insufficient. For example, the material shows significant strength attenuation and insufficient hydrostatic resistance under long-term water immersion conditions; it is susceptible to creep deformation under the action of pressurized water, which leads to a decrease in the seepage control effect; at the same time, the bond strength of the interface between the material and the concrete decreases significantly with time [5,18,19,20,21,22]. To address these problems, this study proposes a new type of modified acrylate grouting material based on the synergistic effect of the composite crosslinking agent. Through the synergistic action of composite crosslinking agents, the comprehensive performance of acrylate grouting materials was significantly improved, and the crosslinking density and toughness of the polymer network were effectively regulated to optimize the mechanical properties and water resistance of the materials. At the same time, through the systematic investigation of the ratio of the curing agent and initiator, the precise control of gel time was realized to meet the needs of different construction environments. In addition, this research has developed a hydrostatic extrusion test method that simulates the actual working conditions, and systematically investigated the service behavior of the material and its mechanism in complex environments. These innovations provide a new theoretical basis and technical path for the development of high-performance acrylate grouting materials, which is of great significance in promoting the progress of seepage and leakage prevention and plugging technology in underground engineering.

## 2. Materials and Methods

### 2.1. Raw Materials

Based on literature reviews and multiple orthogonal proportioning experiments, the main components and the proportional ranges of the grouting material were determined (Figure 1). The selected materials exhibit complementary properties to meet the requirements for mechanical performance, fluidity, and reaction controllability. The specific raw materials and their functional formulations are listed in Table 1.

### 2.2. Sample Preparation

Solution A preparation: Crosslinking agents and triethanolamine were added to the acrylate solution in specified ratios and stirred uniformly.

Solution B preparation: Sodium persulfate was dissolved in deionized water and stirred thoroughly.

Mixing procedure: Solutions A and B were mixed at a mass ratio of 1:1. If the gelation time was too short, a diluted retarder was added to adjust. The preparation process is illustrated in Figure 2.

### 2.3. Performance Testing

Gelation Time Test: The gelation time was measured according to the JC/T 2037–2010 [23] standard for acrylate grouting materials, using the inverted cup method (Figure 3a). The test was conducted under a constant temperature of 25 °C, the uniformly mixed slurry was quickly poured into a standard plastic cup, and then we immediately poured the slurry in the cup into another empty cup of the same specifications. To observe the slurry flow state, repeat the above operation. When the slurry loses its fluidity (i.e., it no longer flows out of the cup), stop the timer and record the total time as the gel time.

Tensile Strength and Fracture Elongation Testing: The tests were conducted in accordance with GB/T 13477.8–2017 [24] (Test Methods for Building Sealants—Part 8: Determination of Tensile Adhesion Properties). Dumbbell-shaped gel specimens (as shown in Figure 3b) were prepared, and the tensile strength and fracture elongation were measured at a curing age of 24 h under standard testing conditions (as illustrated in Figure 3c).

Compressive Strength of Sand Consolidation Test: The test was carried out according to the standard “JC/T 2037–2010”: A metal test mold with an inner diameter of 40 mm and a height of 100 mm was used, filled with standard sand and then filled with a slurry with a gel time control of 20–40 min until it overflowed the surface, and then the mold was dismantled after 24 h of maintenance. Use the universal testing machine to ≤1% per minute deformation rate continuous loading, determine the damage load, and calculate the compressive strength (the damage load and the ratio of the cross-sectional area of the specimen). Six specimens were tested in each group, and the arithmetic average results of the middle four values were taken after eliminating the abnormal values that deviated from the average value by more than 20%, and the test was repeated if there were less than four valid data (Figure 3d).

Viscosity Test: The initial viscosity of the slurry was measured using an NDJ-5S digital viscometer according to the JC/T 2037–2010 standard (Figure 3e).

Water Absorption Test of Gel: The mixed slurry was quickly poured into plastic tubes with a diameter of 10 mm and a length of 50 mm. After curing for 24 h, the samples were demolded. The initial mass *(m*_0_) of the specimens was weighed, and then the specimens were immersed in 500 mL of deionized water. After wiping off the surface water, the mass (*m*_1_) of the specimens was measured at different curing ages, and the water absorption rate (*X*) was calculated using Equation (1) (Figure 3f).(1)$$X=\frac{m_{1}-m_{0}}{m_{0}}\times 100\%$$

Volume Expansion Rate Test of Gel: The initial volume (*V*_0_) of the specimens with a diameter of 10 mm and a length of 50 mm was measured using the water displacement method. After soaking for different curing ages, the volume (*V*) of the gel was measured, and the volume change rate (*W*) was calculated.(2)$$W=\frac{V-V_{0}}{V_{0}}\times 100\%$$

## 3. Results and Discussion

### 3.1. Effect of Acrylate Concentration on Performance

To investigate the influence of acrylate concentration on the performance of grouting materials, the concentration of the acrylate mixture (magnesium acrylate/calcium acrylate/zinc acrylate = 2:2:1) was varied from 20% to 40% while keeping the promoter (TEA) (0.8%), crosslinker (5%), and sodium persulfate (1%) ratios constant. The experimental results are shown in Table 2. As the acrylate mixture concentration increased from 20% to 40%, the initial viscosity of the slurry increased from 2.3 mPa·s to 14.0 mPa·s, the tensile strength of the gel increased from 21 kPa to 88 kPa, the elongation at break increased from 268% to 423%, and the compressive strength of the sand-solidified body increased from 236 kPa to 585 kPa.

From Table 2, it can be observed that the increase in concentration leads to higher slurry viscosity and a comprehensive improvement in mechanical properties. This is because the increase in concentration results in more active components per unit of volume, not only increasing the viscosity of the slurry but also providing more reactive sites, which facilitates the formation of a more complete network structure. From a molecular mechanism perspective, this performance enhancement is mainly due to the reduced average distance between acrylate molecules, which strengthens intermolecular interactions, including van der Waals forces and hydrogen bonding. Additionally, the coordination between divalent metal ions (Mg^2+^, Ca^2+^, Zn^2+^) and carboxyl groups (-COO^−^) is enhanced, promoting the formation of a physical crosslinking network. The increased density of crosslinking points further improves the mechanical strength of the material.

Considering the material properties and engineering applicability, the selection of the acrylate concentration and crosslinking agent dosage needs to balance mechanical properties, permeability, and economic costs. As shown in Table 2, when the acrylate concentration was increased from 30% to 35%, the viscosity of the slurry increased from 5.2 mPa·s to 8.3 mPa·s, and the compressive strength of the sand cement increased from 390 kPa to 440 kPa, while still maintaining good permeability. Although the 40% concentration further improved the mechanical properties (compressive strength of 585 kPa), the slurry viscosity was significantly higher (14.0 mPa·s), permeability was drastically reduced, and the cost of the raw materials increased by about 30%. Therefore, 30–35% was selected as the optimized interval for the acrylate concentration in this study, taking into account both performance and cost-effectiveness.

### 3.2. Effect of Triethanolamine on Gelation Time Under Different Temperature Conditions

To study the influence of temperature and triethanolamine content on the gelation time of acrylate grouting materials, the concentration of acrylate in Solution A was fixed at 35%, the crosslinker was fixed at 5%, and the sodium persulfate in Solution B at was fixed at 1%. The gelation time was investigated at temperatures ranging from 5 °C to 35 °C with varying triethanolamine content amounts. The results are shown in Figure 4. At 5 °C, as the triethanolamine content increased from 0.5% to 2.0% and the gelation time decreased from 1200 s to 280 s. At 35 °C, the gelation time decreased significantly from 360 s to 25 s over the same triethanolamine content range.

The experimental results indicate that both a temperature increase and higher triethanolamine content significantly accelerate the gelation process. According to the analysis of experimental data and the theoretical model, the shortening rate of gel time showed a nonlinear decay characteristic: when the temperature was increased (5 °C→15 °C→25 °C→35 °C), the reduction in gel time caused by the increase in temperature decreased sequentially; similarly, the shortening of gel time decreased sequentially for every 1% increase in the TEA concentration at a fixed temperature. This phenomenon is consistent with the logarithmic decay law implied by the negative coefficients of temperature (T) and concentration (C) in Equation (5), suggesting that there is a decreasing effect of the reaction parameters on the rate regulation. The change in gelation time is mainly regulated by the following mechanisms:

Temperature Increase: Higher temperatures accelerate the thermal decomposition kinetics of sodium persulfate (Na_2_S_2_O_8_), leading to a significant increase in the generation rate of primary radicals (SO_4_^−^•).(3)$${\mathrm{Na}}_{2}S_{2}O_{8}\overset{\Delta }{\to }2{\mathrm{Na}}^{+}+2{{\mathrm{SO}}_{4}}^{-}•$$

Triethanolamine as a Reducing Promoter: Triethanolamine acts as a reducing promoter, undergoing a redox reaction with persulfate ions (S_2_O_8_^2−^), generating secondary radicals R_2_N• with higher reactivity. As the chain radicals grow, crosslinking reactions occur between different chain radicals when crosslinking agents or acrylate molecules capable of crosslinking are present in the system. The crosslinking reactions connect the linear polymer chains, forming a three-dimensional network structure. As the crosslinking reactions proceed, the viscosity of the system gradually increases, ultimately forming a gel.(4)$${S_{2}O_{8}}^{2-}+R_{2}\mathrm{NH}\to {{\mathrm{SO}}_{4}}^{2-}+R_{2}N•$$

Alkaline Nature of Triethanolamine: The alkaline nature of triethanolamine neutralizes acidic by-products generated in the system, maintaining the reaction system at an optimal pH range (6.5–7.5), thereby promoting the efficiency of radical initiation and chain growth reactions.

Through multiple nonlinear regression analysis, a quantitative relationship between gelation time (*t*), absolute temperature (*T*), and triethanolamine concentration (*C*) was established:(5)lgt=3.3−0.036T−0.22C

Experimental data at varying triethanolamine concentrations (1–5%) align well with the theoretical predictions, demonstrating consistent agreement across different initiator doses and temperatures. This model has a high goodness of fit (R^2^ = 0.914), indicating that the empirical equation can effectively predict the variation in gelation time with temperature and concentration. Based on these results, in practical engineering applications, the gelation time can be precisely controlled by adjusting the triethanolamine content and controlling the construction temperature. For example, at room temperature (25 °C), a triethanolamine content of 1.0–1.5% can achieve a suitable gelation time of 60–120 s. In cold environments or when a longer operation time is required, the triethanolamine content can be reduced. In emergency situations requiring rapid water stoppage, the triethanolamine content can be increased to achieve a shorter gelation time.

### 3.3. Effect of Crosslinker Ratio on Mechanical Properties

The crosslinking reaction mechanism determines the formation of the polymer network (e.g., Equation (6), Figure 5):(6)$${\mathrm{nCH}}_{2}=\mathrm{CHOOM}+\left(\mathrm{PEG}500\mathrm{MA},\;\mathrm{PEG}\right)\overset{\mathrm{TEA},\;\mathrm{SPS}}{\to }\mathrm{Crosslinked}\;\mathrm{Acrylate}\;\mathrm{Gel}$$

Among these, PEG500MA and PEG are crosslinkers, SPS is the initiator, and TEA is the accelerator.

The mechanical properties of acrylate grouting materials mainly depend on the regulation of the polymer network structure by crosslinkers. In this experiment, the concentration of acrylate in Solution A was fixed at 35%, the promoter was fixed at 3 parts, the total crosslinker was fixed at 5 parts, and the sodium persulfate in Solution B was fixed at 1%. The effect of different ratios of bifunctional crosslinker PEG500DA to monofunctional crosslinker PEG-MA (ranging from 1:4 to 4:1) on the mechanical properties of the material was investigated. The results are shown in Figure 6.

The experimental results indicate that the ratio of crosslinkers PEG500DA and PEG-MA significantly affects the mechanical properties of the material. As the proportion of PEG500DA increases, the tensile strength of the material increases significantly from 56 kPa to 152 kPa, and the compressive strength of the sand-solidified body increases from 252 kPa to 518 kPa. However, the elongation at break decreases sharply from 610% to 104%. This performance change is mainly attributed to the synergistic mechanism of the two crosslinkers. PEG500DA, as a bifunctional crosslinker, forms a dense three-dimensional network structure with acrylate monomers during polymerization, significantly increasing the number of crosslinking points, thereby enhancing the rigidity and tensile strength of the material. On the other hand, the monofunctional crosslinker PEG-MA provides some sliding space for the molecular chains through its flexible segments, giving the material toughness. As the proportion of PEG500DA increases, the number of crosslinking points increases significantly, restricting the movement of molecular chains, leading to increased strength but reduced ductility. Based on the experimental results, when the PEG500DA/PEG-MA ratio is 2:3, the material exhibits the best balance of strength and toughness, with a tensile strength of 78 kPa, an elongation at break above 407%, and a compressive strength of the sand-solidified body of 336 kPa, meeting the requirements of most engineering applications. It can also avoid the increase in raw material cost and process complexity caused by excessive crosslinking.

### 3.4. Analysis of Factors Affecting Water Absorption and Swelling

To investigate the water stability of modified acrylate gels, the total amount of the crosslinker (3.0%, 4.0%, 5.0%, and 6.0%) was systematically studied under fixed conditions of an acrylate concentration (35%), promoter (3 parts), and the PEG500DA/PEG-MA mass ratio (2:3). The results are shown in Figure 7.

The total amount of the crosslinker significantly affects the water absorption and swelling properties of the modified acrylate gel. Under fixed conditions of an acrylate concentration (35%), promoter (3 parts), and the PEG500DA/PEG-MA mass ratio (2:3), as the total crosslinker content increases from 3.0% to 6.0%, the 28-day water absorption rate of the gel decreases from 36% to 12%, and the volume expansion rate decreases from 32% to 11%. All samples exhibit a rapid water absorption rate and volume expansion in the initial immersion period (0–14 days), followed by a significant slowdown in the rate of increase, stabilizing after 14 days.

This change is mainly attributed to the regulation of the gel network structure by the crosslinker (as shown in Figure 8). The increase in crosslinker content significantly increases the density of chemical crosslinking points, limiting the freedom of molecular chain movement and reducing the average mesh size of the network, thereby increasing the resistance to water molecule diffusion. The experimental results show that when the crosslinker content reaches 5.0% or higher, the water absorption and swelling properties of the material can be effectively controlled, with 28-day water absorption and volume expansion rates below 12% and 11%, respectively, indicating good dimensional stability. This performance optimization is mainly due to the volume expansion of the macromolecular chains combined with the crosslinker after water absorption, resulting in a denser internal network structure of the gel, which hinders the further penetration of water molecules, thereby significantly improving the water stability and long-term anti-seepage performance of the material.

### 3.5. Anti-Extrusion Performance Test

To evaluate the long-term anti-extrusion performance of modified acrylate grouting materials under pressurized water conditions, a set of simulated concrete crack water pressure test devices was designed. The devices consist of two 600 mm × 600 mm × 50 mm concrete plates stacked together, with the crack width controlled between 2 and 5 mm using spacers. The grouting nozzle is installed at the center of the side of the crack, and the interface between the nozzle and the concrete plate is sealed with epoxy resin to ensure that the grout only leaks from the crack (as shown in Figure 9). Due to the limited pressure-bearing capacity of the test mold, G-clamps are installed around the concrete plates to prevent deformation or rupture under water pressure. The samples are immersed in deionized water at 25 ± 2 °C for 1, 3, 7, 14, 28, and 56 days, and then tested according to the SL31–2003 [25] “Water Pressure Test Procedure for Water Conservancy and Hydropower Engineering Drilling.” The pressure is increased stepwise at 0.05 MPa intervals, with each step maintained for 30 min, and the maximum pressure value at which leakage occurs is recorded (as shown in Figure 10).

During the test, a pressure-stabilizing pump was used to slowly apply water pressure, with pressure values increased stepwise at 0.05 MPa (initial pressure), 0.1 MPa, 0.2 MPa, 0.3 MPa, and 0.4 MPa. After each pressure level reached the set value, it was maintained for 30 min, and any extrusion or leakage of the grout was observed. The maximum pressure value that could be reached was recorded. The total duration for a single sample to complete all pressure levels (from 0.05 MPa to 0.4 MPa) was 3.5 h (7 intervals × 30 min each). To simulate actual engineering conditions with pressurized water, the water pressure was kept stable during the test to avoid fluctuations that could affect the results. After the test, the failure mode was observed by splitting the sample, and the leakage path was analyzed using a tracer dye (red pigment) to mark the specific location of the leakage.

In the experiment, the performance of the developed modified acrylate grouting material was compared with two commercially available acrylate grouting materials under the same conditions. The results of the water pressure test at different ages are shown in Figure 11. The experimental results show that the three materials exhibit significant differences under water pressure: Market Sample 1 has a lower initial anti-water pressure strength (about 68 kPa) but relatively stable performance, remaining at around 43 kPa after 56 days. Market Sample 2 has a higher initial anti-water pressure strength (about 133 kPa), but the strength drops sharply after 3 days of immersion, reaching only 20% of the initial value after 56 days. In contrast, the modified acrylate developed in this study exhibits excellent anti-extrusion performance, with an anti-water pressure strength of 300 kPa after 1 day, which continues to increase and stabilizes at around 350 kPa after 56 days, providing long-term water stoppage.

In the water stoppage mechanism of acrylate grouting materials, leakage in the later stage can occur in two failure modes: interface failure and material failure. Using red pigment in the water pressure test, the leakage path is observed after splitting the crack (as shown in Figure 12). The leakage paths of Market Sample 1 and the self-developed sample are both within the gel, indicating good adhesion between the acrylate gel and the concrete interface. However, the leakage path of Market Sample 2 is both within the gel and at the interface with the concrete, indicating the significant performance degradation of the acrylate grouting material, which is prone to re-leakage. In summary, the modified acrylate grouting material developed in this study can effectively cope with complex engineering environments such as high groundwater levels and pressurized water, providing reliable long-term anti-seepage protection for engineering structures, and has significant engineering application value.

### 3.6. Comparative Analysis with Previous Studies

Mechanical Performance Enhancement: Compared with conventional acrylate materials, the compressive strength of the optimized solid sand body reaches 236–585 kPa, which is significantly higher than that of conventional acrylate materials (150–200 kPa) [2] and close to that of polyurethane-based materials (12–24 MPa) [12]. The elongation at break >376%, overcoming the brittleness problem of conventional acrylates (<100%). The comparison is shown in Table 3.Hydraulic Stability under Pressure: In this study, the hydrophobic crosslinking network design is significantly better than the existing materials. In terms of short-term water pressure resistance, the 1-day water pressure resistance reaches 300 kPa, which is better than that of water-soluble polyurethane and cement-based materials; in terms of long-term stability, the 28-day water pressure resistance is stabilized at 350 kPa, which is 75% higher than that of commercial acrylate products (200 kPa), and there is no strength attenuation problem commonly found (a 30% decay rate was reported by Liang et al.) [4]. The viscosity of the slurry, 8.3 mPa·s, combines low viscosity and scour resistance, which is superior to traditional acrylates (10–20 mPa·s) and polyurethanes (solvent dilution required) [6].Durability and Environmental Adaptability: In terms of chemical resistance, the material in this study indirectly improves the resistance to ionic penetration by reducing the water absorption (12%). In terms of environmental protection, the use of polyethylene glycol derivatives (PEG500DA/PEG-MA) avoids the toxicity of organic solvents in the case of epoxy resins and the problem of the volatilization of isocyanates in the case of polyurethanes.

## 4. Conclusions and Outlook

### 4.1. Key Findings

This study presents a composite crosslinking strategy to enhance the mechanical strength, hydraulic stability, and engineering applicability of acrylate grouting materials. The main advancements are summarized as follows:Material Performance Optimization and Concentration Regulation: Using a mixed solution of magnesium, calcium, and zinc acrylate as the base, combined with the synergistic effect of a bifunctional crosslinker (PEG500DA) and a monofunctional crosslinker (D), the mechanical properties and water stability of the material were significantly improved. When the acrylate concentration was increased to 35%, the slurry viscosity (8.3 mPa·s) and mechanical properties (tensile strength of 76 kPa, compressive strength of sand-solidified body of 440 kPa) reached an optimal balance, while maintaining excellent permeability, meeting the needs of complex engineering environments.Precise Control of Gelation Time: By adjusting the triethanolamine content and construction temperature, the gelation time can be flexibly controlled. At room temperature (25 °C), a triethanolamine content of 2–3% can stabilize the gelation time at 60–120 s, while under high-temperature or high-content conditions, it can be further shortened to 25 s, providing adaptive solutions for different construction scenarios (such as emergency water stoppage or low-temperature environments).Crosslinker Ratio and Water Stability: The ratio of crosslinkers PEG500DA to D directly affects the strength–toughness balance of the material. When the ratio is 2:3, the material exhibits both high tensile strength (78 kPa) and excellent ductility (elongation at break >407%). At the same time, when the total crosslinker content reaches 5%, the 28-day water absorption rate (12%) and volume expansion rate (11%) are significantly reduced, indicating that the material can maintain a stable structure under long-term immersion.Breakthrough in Anti-Water Extrusion Performance: Through simulated pressurized water environment tests, the anti-water pressure strength of the modified acrylate grouting material reached 300 kPa after 1 day and stabilized at 350 kPa after 28 days, far exceeding that of commercially available products (<200 kPa). The failure mode is mainly internal gel failure, with excellent interfacial bonding strength, effectively resisting the risk of leakage under high-water-pressure environments, providing reliable long-term anti-seepage and water stoppage protection for underground engineering.

In summary, the modified acrylate grouting material developed in this study, through composite crosslinking strategies and formulation optimization, achieves synergistic improvement in high-strength, water resistance, and anti-water pressure performance, solving the problem of the performance degradation of traditional materials under complex working conditions, and has significant engineering application value.

### 4.2. Study Limitations

Despite these advancements, this work has three key limitations:Lack of Microstructural Characterization: The polymer network’s crystallinity, crosslinking density, and interfacial bonding mechanisms (e.g., gel–aggregate interactions) were not analyzed using XRD, SEM-EDS, or FTIR. This omission hinders a mechanistic understanding of how crosslinker ratios regulate chain arrangements and phase-interface properties.Narrowed Environmental Testing: While hydraulic stability was validated under controlled laboratory conditions, the material’s performance in extreme environments (e.g., high salinity, dynamic water flow) remains untested, limiting direct extrapolation to field applications.Limited Multi-Factor Interaction Analysis: The single-factor experimental design identified dominant trends but did not systematically quantify synergistic or antagonistic effects between components, potentially underestimating optimization potential.

### 4.3. Future Directions

To address these gaps and extend the current work, the following steps are proposed:Microstructural Analysis: Employ XRD to quantify crosslinker-induced crystallinity changes and polymer chain alignment. Use SEM-EDS to map elemental distributions at gel–aggregate interfaces, elucidating bonding enhancement mechanisms. Apply in situ FTIR to track chemical bond evolution during polymerization, refining reaction kinetic models.Expanded Environmental Testing: Conduct field trials under extreme hydrogeological conditions (e.g., saline groundwater and a turbulent flow) to validate long-term stability and adaptability.Multi-Component Interaction Modeling: Apply response surface methodology (RSM) coupled with genetic algorithms (GAs) to resolve nonlinear interactions among key variables (e.g., the acrylate concentration, the crosslinker ratio, and the initiator/promoter content).Material Optimization: Explore ternary crosslinking systems or hybrid organic–inorganic networks to further balance strength and ductility while reducing production costs.

## Figures and Tables

**Figure 1 polymers-17-00827-f001:**
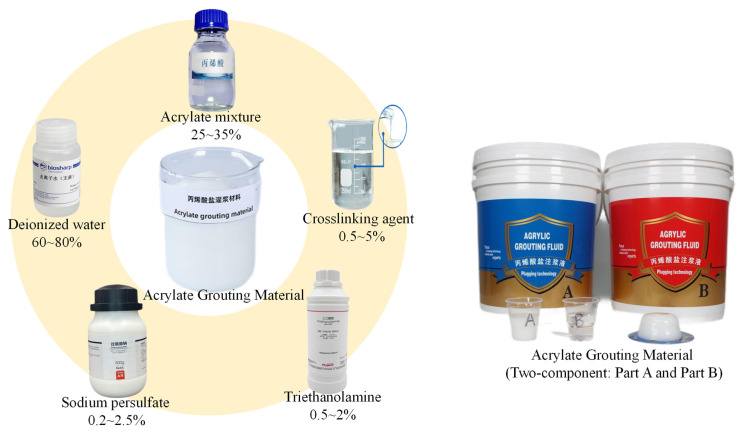
Schematic diagram of acrylate composition and content.

**Figure 2 polymers-17-00827-f002:**
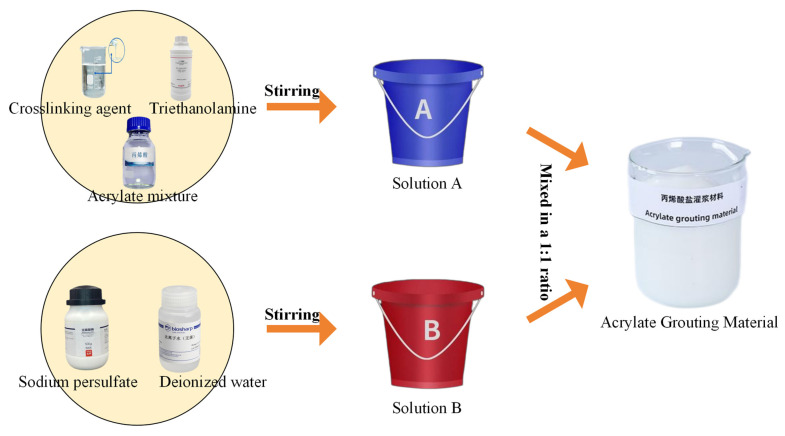
Schematic flowchart of acrylate grouting material preparation.

**Figure 3 polymers-17-00827-f003:**
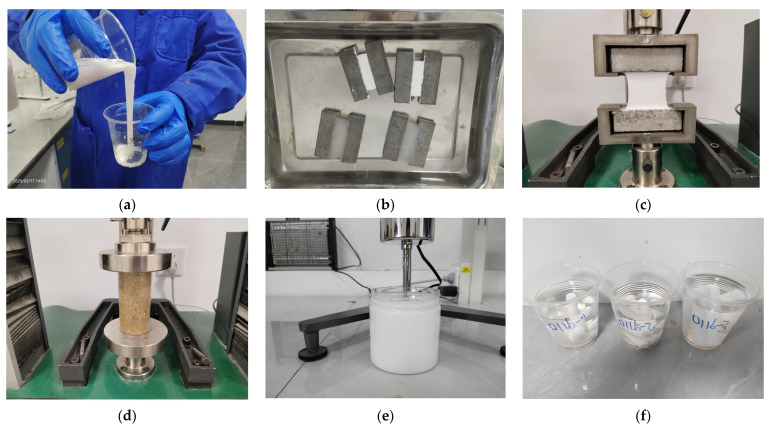
Performance testing methods: (**a**) inverted cup method, (**b**) mortar plate specimen, (**c**) Tensile Test, (**d**) Compression Test, (**e**) Viscosity Test, (**f**) Water Absorption and Swelling Test.

**Figure 4 polymers-17-00827-f004:**
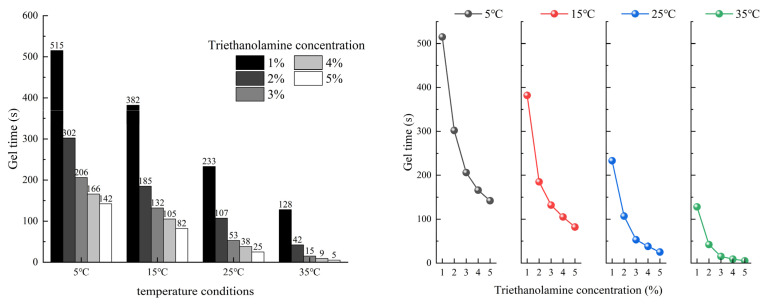
Effect of triethanolamine content and temperature on gelation time.

**Figure 5 polymers-17-00827-f005:**
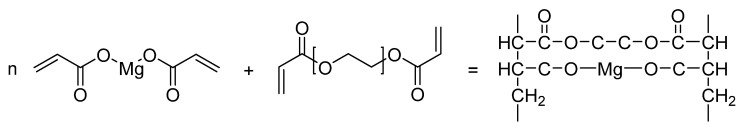
Reaction mechanism diagram of crosslinked acrylate gel.

**Figure 6 polymers-17-00827-f006:**
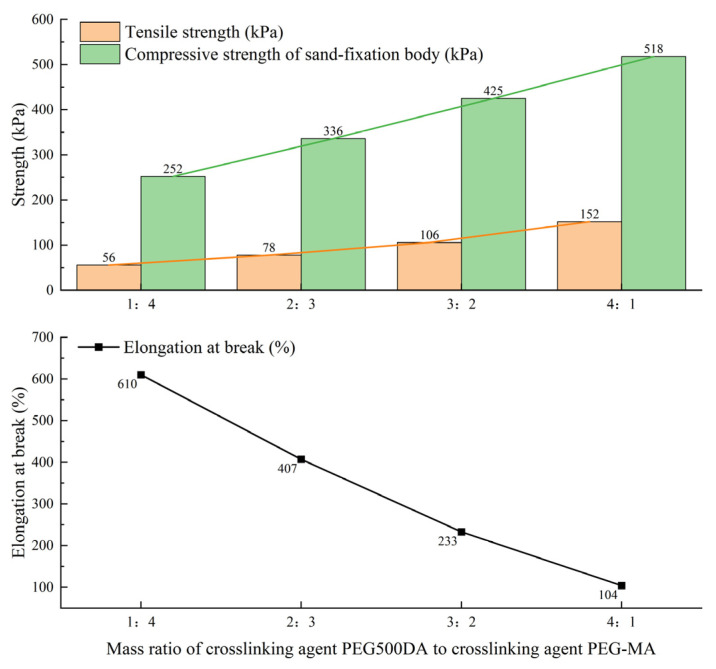
Effect of different ratios of two crosslinkers on material performance.

**Figure 7 polymers-17-00827-f007:**
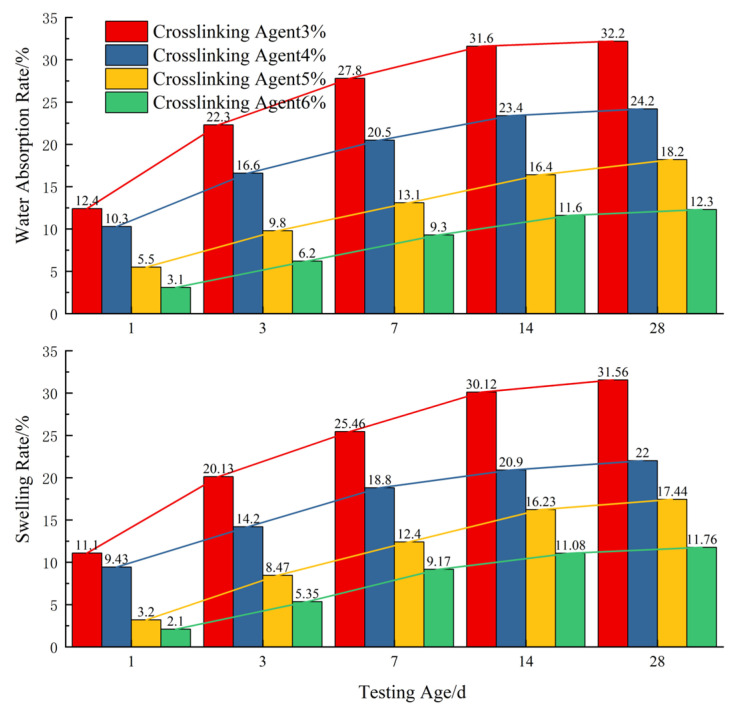
Water absorption and volume expansion rates of gel at different crosslinker contents over time.

**Figure 8 polymers-17-00827-f008:**
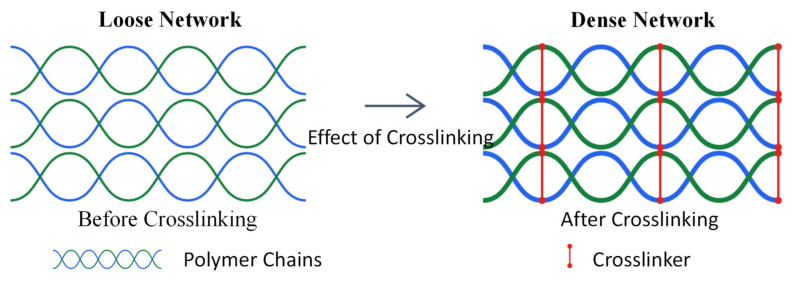
Schematic diagram of gel network structure before and after crosslinking.

**Figure 9 polymers-17-00827-f009:**
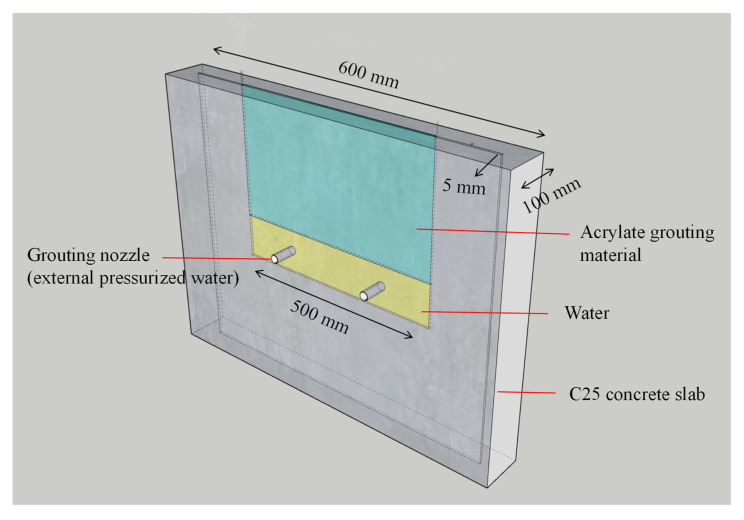
Water pressure test model.

**Figure 10 polymers-17-00827-f010:**
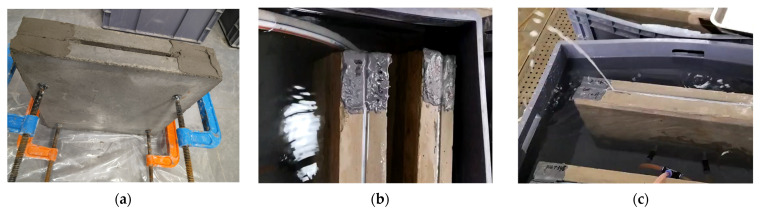
Water pressure test procedure: (**a**) sample preparation, (**b**) sample immersion in water for curing, (**c**) water pressure test.

**Figure 11 polymers-17-00827-f011:**
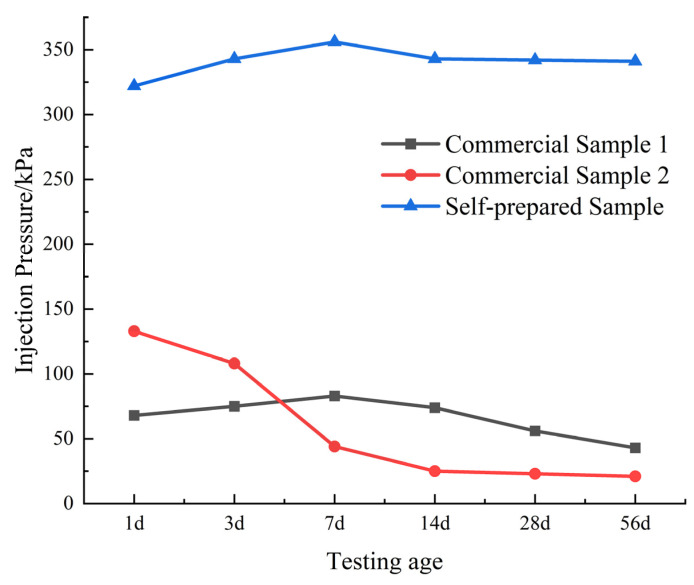
Evaluation of water extrusion resistance of gels at different ages.

**Figure 12 polymers-17-00827-f012:**
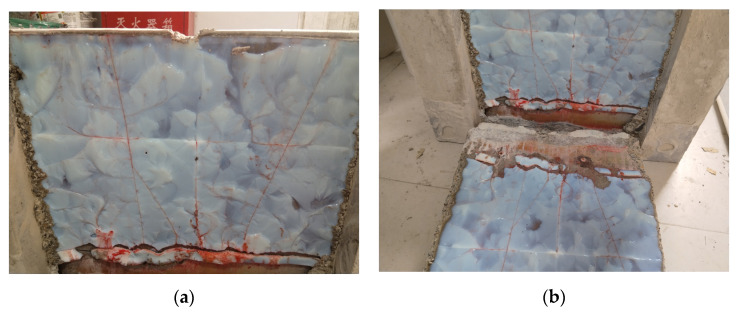
Leakage path diagram: (**a**) inside the gel, (**b**) interface.

**Table 1 polymers-17-00827-t001:** Components and proportions of modified acrylate grouting materials.

Material Composition	Weight Ratio (%)	Functional Description	Functional Classification	Manufacturing Company	Fineness
Acrylate mixture	25–35	A blend of magnesium acrylate, zinc acrylate, and calcium acrylate in a specific ratio, serving as the primary matrix for bonding and mechanical performance.	Matrix material	Oriental Yuhong Waterproof Technology Co., Ltd. (Beijing, China)	95–99
PEG diacrylate (PEG500DA)	0.5–5	Bifunctional crosslinking agent that enhances intermolecular crosslinking density, improving mechanical strength and structural stability.	Crosslinking agent	Changxing Chemical (Qingdao, China)	98–99
PEG monoacrylate (PEG-MA)	0.5–5	Monofunctional crosslinking agent that regulates crosslinking degree to optimize flexibility and processability.	Crosslinking agent	Self-restraint	95–98
Triethanolamine (TEA)	0.5–2	Catalyzes polymerization and adjusts pH to ensure uniform and stable reaction conditions.	Curing accelerator/Promoter	Dow Chemical Company (Shanghai, China)	99
Sodium persulfate (SPS)	0.2–2.5	Initiates free radical polymerization of acrylate monomers to promote polymer chain formation.	Polymerization initiator	Langfang Pengcai Fine Chemical Co. (Langfang, China)	98–99
Potassium ferricyanide	0–0.1	Inhibits polymerization rate to prevent rapid reaction-induced inhomogeneity.	Reaction regulator	Commercially Available	99
Deionized water	60–80	Adjusts system concentration and fluidity for uniform dispersion and reaction efficiency.	Solvent	Commercially Available	Conductivity <1 μS/cm

**Table 2 polymers-17-00827-t002:** Effect of acrylate concentration on performance.

Acrylate Concentration	20%	25%	30%	35%	40%
Initial Viscosity (mPa·s)	2.3	3.6	5.2	8.3	14
Tensile Strength (kPa)	21	38	62	76	88
Compressive Strength of Sand-Solidified Body (kPa)	236	325	390	440	585
Elongation at Break (%)	268	322	376	405	423

**Table 3 polymers-17-00827-t003:** Performance comparison of modified acrylate composite vs. conventional grouting materials.

Performance Indicators	Data for This Study	Conventional Acrylics	Polyurethane Materials	Cementitious Materials (Class IV)
Compressive strength of solid sand	385–440 kPa	150–200 kPa	12–24 MPa	65–85 MPa
Elongation at break	>320%	100%	1000–2000%	<0.5%
Absorption rate of 28 days of water	12%	25–30% (Long-term water absorption)	<5%	15–20%
Resistance to water pressure (28 days)	350 kPa	<200 kPa	≥0.8 MPa	impermeability rating ≥ P8

## Data Availability

Data are contained within the article.

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
