# Peer review of "Optimization of Water Plugging Characteristics and Mechanical Properties of Acrylate Grouting Materials Based on Composite Crosslinking Strategy"

_polymers, 2025, doi:10.3390/polym17060827_

Round 1
Reviewer 1 Report
Comments and Suggestions for Authors
1) This study evaluated the mechanical properties and water stability of the composites using the modified acrylates and crosslinking agents for grouting materials. The work is good, and the results have been analysed systematically. The topic is both interesting and important, but the current manuscript lacks a description comparing your experimental results with those of other researchers. As the authors mention in the Introduction, many reports have detailed the use of various acrylates and crosslinking agents in conjunction with grouting materials.
2) The macro-level evaluation focused on water plugging, ductility, tensile strength, compressive strength, elongation break, and bonding properties, while the microstructural analysis of the acrylate grouting material at its optimal mix ratio did not utilize techniques such as XRD, SEM-EDS, and infrared spectroscopy.
3) In page 4, lines 109-110 (in The Sample Preparation section); The authors said that “Mixing procedure: Solutions A and B were mixed at a mass ratio of 1:1”. Why was a 1:1 mass ratio preferred? Wouldn't it be better to do it in 1:4, 1:2, and 3:4 mass ratios?
4) In page 8, lines 247-251; The authors said that “Based on the experimental results, when the PEG500DA/D ratio is 2:3, the material exhibits the best balance of strength and toughness, with a tensile strength of 95 kPa, elongation at break above 320%, and a compressive strength of the sand-solidified body of 385 kPa, meeting the requirements of most engineering applications”. On the other hand, when Figure 6 is examined, it is seen that when the mass ratio of the crosslinking agent is 2:3, the tensile strength is 78 kPa, the elongation at break is 336% and the compressive strength is 407 kPa. There seems to be a contradiction here. The authors should clarify or correct it.
5) The numerous acrylates (concentrations up to 40%) and crosslinking agents (amounts up to 5%) are required to make the process very expensive. The authors do not adequately discuss the quantities of acrylates and crosslinking agents used and the costs required.
6) The Conclusion is unsatisfactory. It is recommended that the “Conclusion” section be rewritten. There is no “Discussion” section at all!
7) The authors summarized the results of the experiments in the Conclusion section. However, the study's shortcomings, what their future studies would be, and their suggestions and opinions to other researchers were never mentioned.
Considering the above feedback, I think publishing the present article in the Polymers journal with significant revisions is appropriate.
Comments on the Quality of English Language8) The English appears to be quite good. However, it would be useful to review the grammar of the existing manuscript.
Reviewer 2 Report
Comments and Suggestions for Authors
The manuscript focuses on the development of acrylate grouting materials for waterproofing, with improved properties achieved through a comprehensive study of the effects of various component concentrations on the integral properties of the modified material. This study is primarily applied in nature and holds significant practical value in the modern world. However, the scientific novelty of the manuscript appears to be somewhat limited. Nevertheless, this does not diminish its overall quality. The manuscript is engaging to read, and the results obtained can serve as a foundation for numerous subsequent studies.
Before publication, the authors should carefully address certain weaknesses of the manuscript:
- The structure of the study appears somewhat unusual. The investigated system comprises at least five components, yet each section examines the correlation between one specific physical property of the composition and the variation in the concentration of a single component. However, it is evident that changes in the concentrations of other components will also influence the physical parameters of the mixture. The authors should provide a justification for the chosen structure of the study and explain why they specifically investigate the effect of a particular component on a given physical property (for example, the effect of the crosslinking agent on gel volume change and water absorption).
- How was the completeness of the gelation process controlled? It is well known that crosslinked systems continue to develop an entanglement network after losing fluidity, which can significantly alter their properties, particularly mechanical characteristics. Did the authors investigate how the gel properties change over different aging times?
Below is a list of minor comments that also require revision:
- Section 2.1: Raw Materials – The authors should specify the purity of the materials and their manufacturers.
- The abbreviations used for Sodium Persulfate (DSP) and PEG monoacrylate (D) should be revised to conform to standard scientific notation or be made more intuitive to facilitate readability. The current abbreviations require constant reference back to their definitions. The authors may consider using more conventional notations such as “PEG-ma” or others at their discretion.
- Line 115: Information regarding standard JC/T 116 2037-2010 is difficult to obtain. The authors are advised to briefly describe the testing procedure and clarify how identical conditions were maintained across different samples. Additionally, why were standard rheological tests not used to study gelation?
- Figure 3d: Was the sand impregnated with the investigated composition? How was the impregnation performed, and how was the completeness of pore filling evaluated?
- Line 147: The term "Promoter" should be clarified or defined.
- Lines 166-168: The authors should substantiate the claim that these parameters are indeed optimal, for instance, by referencing regulatory documents or previous studies. Otherwise, it remains unclear on what basis this conclusion was drawn. Additionally, 14 mPa·s corresponds to a low-viscosity suspension. Since these systems are intended for application, surface tension is also a critical parameter, as the compositions must wet narrow gaps and penetrate pores. The authors should consider investigating the surface tension of the prepared mixtures.
- Figure 4: The figure is difficult to interpret. Since time is a key factor, it should be clearly indicated on the graph. The triethanolamine dosage is also not informative—its concentration should be provided. It is strongly recommended that the authors revise this figure and consider splitting it into 2-3 separate graphs to present in subsequent sections (Temperature Increase, Triethanolamine as a Reducing Promoter, Alkaline Nature of Triethanolamine). Furthermore, a correlation should be established between reaction parameter changes and reaction rate.
- Lines 198-202: An illustration comparing the obtained data with the theoretical dependency for different initiator doses is required.
- Lines 211-224: This section repeats information already presented in the introduction and in Section 2: Materials and Methods.
- In Section 3.3, why was the 1:1 component ratio not examined? From an application standpoint, such a ratio would be more convenient and, judging by the data, might not perform worse than the 2:3 ratio for engineering applications.
- Section 3.5: Anti-Extrusion Performance Test – There appears to be a typographical error in the text, which states that the pressure was increased gradually by 0.05 MPa/30 min, reaching a maximum pressure of over 300 MPa. How long was the test conducted under this regime? What was the initial pressure?
- Figure 9 should be enlarged. The geometric dimensions should be labeled, and the materials used should be clearly identified, as the current figure lacks clarity.
Addressing these comments will significantly improve the clarity and overall quality of the manuscript.
Round 2
Reviewer 1 Report
Comments and Suggestions for Authors
With the authors making the necessary edits to my criticisms on the current manuscript, I think the quality of the paper became suitable for publication in this prestigious journal. Therefore my suggestion will be acceptance in its present form.